# Rapid Determination of Mixed Pesticide Residues on Apple Surfaces by Surface-Enhanced Raman Spectroscopy

**DOI:** 10.3390/foods11081089

**Published:** 2022-04-10

**Authors:** Luyao Wang, Pei Ma, Hui Chen, Min Chang, Ping Lu, Ning Chen, Yanbing Yuan, Nan Chen, Xuedian Zhang

**Affiliations:** 1Key Laboratory of Optical Technology and Instrument for Medicine, Ministry of Education, College of Optical-Electrical and Computer Engineering, University of Shanghai for Science and Technology, Shanghai 200093, China; wangluyao0106@126.com (L.W.); peima@usst.edu.cn (P.M.); chenhui@usst.edu.cn (H.C.); changmin@usst.edu.cn (M.C.); lu945632952@163.com (P.L.); chenning103420@163.com (N.C.); 15054596156@163.com (Y.Y.); 2School of Electrical Engineering, Nantong University, Nantong 226019, China; ntu_chennan@ntu.edu.cn; 3Shanghai Institute of Intelligent Science and Technology, Tongji University, Shanghai 200092, China

**Keywords:** SERS, pesticide, CPF, 2,4-D, nanoparticles, apple

## Abstract

Chlorpyrifos (CPF) and 2,4-dichlorophenoxyacetic acid (2,4-D) are insecticides and herbicides which has been widely used on farms. However, CPF and 2,4-D residues on corps can bring high risks to human health. Accurate detection of pesticide residues is important for controlling health risks caused by CPF and 2,4-D. Therefore, we developed a fast, sensitive, economical, and lossless surface-enhanced Raman spectroscopy (SERS)-based method for pesticide detection. It can rapidly and simultaneously determine the CPF and 2,4-D mixed pesticide residues on an apple surface at a minimum of 0.001 mg L^−1^ concentration, which is far below the pesticide residue standard in China and the EU. The limits of detection reach down to 1.28 × 10^−9^ mol L^−1^ for CPF and 2.47 × 10^−10^ mol L^−1^ for 2,4-D. The limits of quantification are 4.27 × 10^−9^ mol L^−1^ and 8.23 × 10^−10^ mol L^−1^ for CPF and 2,4-D. This method has a great potential for the accurate detection of pesticide residues, and may be applied to other fields of agricultural products and food industry.

## 1. Introduction

CPF and 2,4-D are widely used insecticides and herbicides in the world, and have proven to be an effective pest control method in different varieties of fruits and vegetables [1,2]. Among these, CPF, an organophosphate insecticide [3] has been one of the most-selling insecticides. However, CPF residues may accumulate in the body and cause a variety of diseases. It can also affect neurological development of children [4,5,6]. 2,4-D is a widely-used herbicide and plant growth regulator in agriculture all over the world, and the application of 2,4-D has continued to increase in recent years [7,8]. Moreover, the residues of 2,4-D dose in the body may contribute to the development of Alzheimer’s disease and different cancer [9]. The mixed application of pesticides has been very common, especially the combination of an insecticide and an herbicide [10]. Mixed application of pesticide residues will bring more serious and complicated risks to human health [11]. Obviously, accurate and convenient detection of pesticide residues is very important in food risk control.

Apple is a widely cultivated fruit tree and an economic tree in temperate regions of the world [12]. In addition, apple’s polyphenols are thought to be beneficial for arterial blood pressure and hyperlipidemia. However, in order to protect crop yields and reduce losses, they are often exposed to pesticides [13].

At present, many methods are applied to the detection of pesticide residues in food, such as liquid chromatography-tandem mass spectroscopy (LC-MS/MS) [14], gas chromatography–mass spectrometry (GC-MS) [15,16], gas chromatography (GC) and liquid chromatography (LC) quadrupole-time-of-flight mass spectrometry (Q-TOFMS) [17], ultra-performance liquid chromatography, coupled to tandem mass spectrometry (UHPLC–MS/MS) [18], and ambient ionization tandem mass spectrometry [19]. Although these methods are reliable, sensitive, and stable, the complex operation, requirements of special working environment, time-consuming operation time and special storage conditions of samples bring challenges to operators and samples. Simple, rapid, and convenient detection methods of pesticides, especially mixed pesticides are still in a great need in agricultural fields.

Raman spectroscopy has attracted much attention as it is a nondestructive analytical technique that can provide detailed information about the chemical structure, crystallinity and molecular interactions of samples [20]. The surface-enhanced Raman spectroscopy (SERS) overcomes the weakness of the conventional Raman spectroscopy, which is enough to detect the Raman signal of a single molecule [21,22]. Recently, SERS has become a reliable technique and been widely used to detect trace pesticide residues in food [23,24,25,26,27,28]. SERS was used in the detection of three organophosphorus pesticides with gold nanoparticle [29]. Fabricated micro-bowl array SERS substrate was employed to detect pesticide residue on vegetables [30]. SERS can also determine the chlorpyrifos in tomato [31]. In actual agricultural production, two or more types of pesticides are often mixed to meet production requirements, SERS has shown a great potential in determining the content of mixed pesticides [32,33].

This study aimed to evaluate and establish a method to determination CPF and 2,4-D residue on apples surfaces by SERS. In our study, silver colloid was used as the enhancing base for the detection of mixed pesticide residues on apples surfaces. The SERS characteristic peaks of two kinds of pesticides were selected to establish linear fitting equation to realize qualitative and quantitative detection. The detection limit of this method in mixed pesticide detection can reach 10^−9^ mol L^−1^, and this method may be applied for the rapid detection and analysis of different pesticides in agricultural products. In addition, this method is simple and time-saving, therefore it is promising in becoming a standard analytical tool.

## 2. Materials and Methods

### 2.1. Synthesis of Silver Colloid

Silver colloid, was prepared by reduction of silver nitrate with hydroxylamine hydrochloride at alkaline pH and at room temperature [34]. Typically, 10 mL of mixed solution containing 1.5 × 10^−2^ mol L^−1^ Hydroxylammonium chloride (H_3_NOHCl, Macklin Reagent, Shanghai, China) and 3 × 10^−2^ mol L^−1^ Sodium hydroxide (NaOH, Macklin Reagent, Shanghai, China) solution was rapidly added to 90 mL of silver nitrate solution (AgNO_3_, 10^−3^ mol L^−1^, Carbon Twelve Reagent, Shenzhen, China) under vigorous stirring. After the solution color turned to gray, the silver colloid was obtained and stored in dark in an amber bottle.

Silver colloid were characterized by an Evolution 350 UV-Vis Spectrophotometer (Thermo Scientific, Shanghai, China), and Figure 1a shows the UV–visible absorption spectra in the range of 300–800 nm. The result exhibited that the silver nanoparticles UV/Vis absorption band at 409 nm and the smooth spectral curve. Transmission electron microscopy (TEM) images were obtained with a JEM-2100F (JEOL, Tokyo, Japan), electron microscope operating at 200 kV, and the nanoparticle size distribution was measured with dynamic light scattering experiments (Zetasizer, Nano-ZS90, Malvern, UK). The dynamic light scattering and TEM images showed that the size of silver nanoparticles was consistent and there was no serious agglomeration. The particle size of the prepared silver colloid is 81 nm.

### 2.2. Standard Solution Preparation

CPF standard solution, 2,4-D standard solution, and mixed standard solution (CPF and 2,4-D concentration ratio C:C=1) were prepared using methanol (Aladdin Reagent, Shanghai, China) and ultrapure water (18.25 MΩ) (methanol/ultrapure water = 1:1, *v/v*)) [31]. According to the GB 2763-2019, CPF and 2,4-D in apples should not exceed 1 mg kg^−1^ (EU—0.01 mg kg^−1^) and 0.01 mg kg^−1^ (EU-0.05 mg kg^−1^), respectively. Therefore, we prepared seven concentrations (1000, 100, 10, 1, 0.1, 0.01, 0.001 mg L^−1^) CPF solutions, eight concentrations (1000, 100, 10, 1, 0.1, 0.01, 0.001, 0.0001 mg L^−1^) 2,4-D solutions, and seven concentrations (1000, 100, 10, 1, 0.1, 0.01, 0.001 mg L^−1^) mixed solutions. CPF and 2,4-D at a concentration of 1 mg L^−1^ were respectively by volume ratio mixed preparation (1:1, 1:2, 1:3, 1:4, 1:5, 1:6, 6:1, 5:1, 4:1, 3:1, 2:1, 1:1 CPF/2,4-D).

### 2.3. Sample Preparation

Organic Red Fuji Apples were purchased from a local supermarket in Shanghai. Referring to the previous experimental method [31], the apples were first washed with methanol and deionized water several times and placed in room temperature for drying [32,35,36]. Next, a piece of apple skin was cut and spread on the glass slide, and pesticide of a certain concentration was added. After the solution was dried at room temperature, silver colloid was added to cover the surface, and then used for SERS detection when it was close to drying. The procedures were illustrated in Figure 2.

### 2.4. SERS Spectral Collection and Data Analysis

Throughout the study, SERS spectra were obtained using a commercial Raman system (LabRAM Xplora Plus, HORIBA Scientific, Paris, France). Raman spectra ranging from 300 to 1800 cm^−1^ were recorded with an incident laser wavelength of 638 nm and a 10× objective lens. The typical accumulation time used in this study was 2 s [37]. The obtained Raman spectra and SERS spectra were preprocessed. First, the cosmic ray effect was eliminated and each spectrum was smoothed to reduce the influence of noise. Moreover, polynomial fitting was used for baseline correction to eliminate the influence of fluorescence background [38,39] with the order set to 8. Both operations were performed in LabSpec6 software (HORIBA Scientific, Paris, France, 2017).

The correlation between the intensity of the characteristic peaks of SERS spectra and the concentration of the solution was analyzed, and a linear relationship between the logarithm of the characteristic peaks intensity (log I) of the sample and the logarithm of the sample concentration (log C) was established. When the concentration range is large, the linear relationship between the logarithmic peak intensity and the logarithmic concentration is usually adopted [40]. Each data point obtained the calculation result of intensity by analyzing the SERS spectrum of six independent measurements, and the error bar gave the standard deviation (SD) of these six I values under a certain concentration of sample. The SD is calculated by the formula of$\;\mathrm{SD}=\sqrt{\sum_{i=1}^{n}I_{i}-\bar{I}/n-1}$ and $\bar{I}=\left(1/n\right)\sum_{i=1}^{n}I_{i}$ is the mean of the six I values (*n* = 6). Linear equation was established for quantization, and R^2^ value determined the degree of fitting. The relative standard deviation (RSD) was also calculated to obtain the measurement accuracy, which are reported in the Appendix A [41,42,43].

The prediction accuracy of the model was evaluated in terms of the recovery rate, which was calculated as the ratio of the average predicted value to the actual value for each concentration [44]. The average predicted concentrations were obtained by averaging the six spectral intensities for each concentration and put into the linear calibration curve.

## 3. Results and Discussion

### 3.1. Spectral Features of CPF and 2,4-D Solid Sample

Figure 3 shows the molecular geometry and Raman spectra of CPF and 2,4-D, respectively. The measurement results of the two powders agree with those described in the literature [16,45,46,47]. The main Raman characteristic peaks of CPF, such as 341 cm^−1^, 613 cm^−1^ and 675 cm^−1^ were both detected, as well as the main Raman characteristic peaks of 2,4-D, 392 cm^−1^, 855 cm^−1^, and 1590 cm^−1^. In addition, some characteristic peaks of weak signals were detected. More detailed characteristic peaks are shown in Table 1.

### 3.2. SERS Measurement and Analysis of CPF and 2,4-D Standard Solutions

Different concentrations of CPF, 2,4-D and mixed pesticide standard solution SERS detection results were shown in Figure 4. Figure 4a,b exhibits the SERS spectra of CPF and 2,4-D at different concentrations, respectively. The characteristic peaks of CPF were observed at 341 cm^−1^, 613 cm^−1^, 675 cm^−1^, 1269 cm^−1^, and 1567 cm^−1^ and those of 2,4-D were observed at 392 cm^−1^, 855 cm^−1^, 945 cm^−1^, 1101 cm^−1^, and 1415 cm^−1^ in the SERS spectra, and the corresponding vibration attribution is summarized in Table 1. The spectral intensities of both CPF and 2,4-D were positively correlated with pesticide concentrations. When the CPF standard solution concentration was 0.001 mg L^−1^ (2.85 × 10^−9^ mol L^−1^), the characteristic peaks of 341, 613, and 675 cm^−1^ can be clearly observed. Moreover, when 2,4-D standard solution concentration was 0.0001 mg L^−1^ (4.5 × 10^−10^ mol L^−1^), which can still be observed. The limits of detection reach down to 1.28 × 10^−9^ mol L^−1^ for CPF and 2.47 × 10^−10^ mol L^−1^ for 2,4-D by 3σ/s method, based on the calculations in the previous study [31,48].

More Raman characteristic peaks of CPF and 2,4-D standard solutions were listed in Table 1. In the process of SERS detection, some of the characteristic peaks had slight Raman frequency shift, which was due to the correlation between molecular chemical bonds and vibration modes [49].

The SERS spectra of 1:1 mixed CPF and 2,4-D solution were shown in Figure 4c,d. Figure 4c showed the SERS spectrum of the standard solution of CPF, 2,4-D and their mixed solution. In the mixed standard sample, the respective Raman characteristic peaks of CPF and 2,4-D can be obviously observed. The characteristic peak strength changes with the concentration gradient of the mixed pesticides as shown in Figure 4d. It is obvious that the characteristic peaks of CPF and 2,4-D at 0.001 mg L^−1^ could still be detected, such as 341 cm^−1^ for CPF and 392 cm^−1^, 613 cm^−1^ and 675 cm^−1^ for 2,4-D [46,50,51]. By observing the SERS spectrum of the mixed solution, it was found that the characteristic peaks of CPF and 2,4-D had hardly shifted, demonstrating the feasibility of SERS detection for mixed pesticides.

Table 2 showed the linear equations between log I and log C at the three characteristic peaks of CPF (342, 613, and 675 cm^−1^) and 2,4-D (392, 852, and 1587 cm^−1^), respectively. R^2^ value indicated optimum fitting results. The characteristic peak of CPF at 675 cm^−1^ had the highest R^2^ value (0.97) and the characteristic peak of 2,4-D at 392 cm^−1^ had the highest R^2^ value (0.98).

The relationship between the intensity of the characteristic peak and the concentration of the sample was shown in Figure 5. Figure 5a,c shows the linear coordinates relationships of intensities of the SERS characteristic peak at 675 cm^−1^ and 392 cm^−1^ and concentrations of CPF and 2,4-D, respectively. Figure 5b,d shows the log coordinates relationships of intensities of the SERS characteristic peak at 675 cm^−1^ and 392 cm^−1^ and concentrations of CPF and 2,4-D, respectively.

### 3.3. SERS Measurements and Analysis of Apple Surface Samples Treated with CPF and 2,4-D

In this study, CPF and 2,4-D were used as examples to detect individual/mixed pesticides in apple surface. Raman spectra of pesticide residues on apple surface were obtained using the previous method coupled with the SERS (Figure 2). The apple surface was treated by CPF and 2,4-D respectively, and the spectra were measured as shown in Figure 6a,b. The Raman characteristic peak at 341, 613, 675, 1094, and 1567 cm^−1^ could be observed [50,51,52]. In the same way, the apple surface treated with 2,4-D was detected, and the Raman characteristic peaks at 392, 696, 855, 945, and 1590 cm^−1^ were observed. The results of apple surface samples treated with CPF and apple surface samples treated with 2,4-D were consistent with those of standard samples. The Raman characteristic peak of the spectra showed a strong positive correlation with the concentration of the pesticide. When the apple surface samples treated with CPF was 0.001 mg L^−1^, the characteristic peaks of 341, 613, and 675 cm^−1^ can be clearly observed. Moreover, when apple surface samples treated with 2,4-D was 0.0001 mg L^−1^, 392, 855, and 1590 cm^−1^ can still be observed. The minimum detection concentrations are well below the maximum residue levels set by the National Food Safety Standard of China (1 mg kg^−1^ for CPF and 0.01 mg kg^−1^ for 2,4-D) and EU (0.01 mg kg^−1^ for CPF and 0.05 mg kg^−1^ for 2,4-D).

Table 3 gives unary linear fitting equations for log I and log C of CPF at 341, 621, and 675 cm^−1^, and 2,4-D at 392, 852 and 1587 cm^−1^. By comparing the R^2^ values, it was found that the Raman characteristic peaks at 675 cm^−1^ and 392 cm^−1^ had the best fitting. The relationship between the intensity of the characteristic peak and the concentration of the sample was shown in Figure 7. Figure 7a, c shows the linear coordinates relationships of intensities of the SERS characteristic peak at 675 cm^−1^ and 392 cm^−1^ and concentrations of CPF and 2,4-D, respectively. Figure 7b, d shows the log coordinates relationships of intensities of the SERS characteristic peak at 675 cm^−1^ and 392 cm^−1^ and concentrations of CPF and 2,4-D, respectively.

To verify the accuracy of SERS measurement, we calculated the relative standard deviation (RSD) of concentration gradient between CPF and 2,4-D standard solution and apple epidermis sample, which are reported in the Appendix A. In CPF, 2,4-D standard solution and apple surface sample, RSD value increased gradually with the decrease in solution concentration, but it was still lower than 10%, which was considered reasonable in SERS measurement [42,44]. The results show that the silver colloid can be used to detect CPF and 2,4-D with high sensitivity and sufficient accuracy. The standard concentration, detected concentration and recovery rate of each sample are shown in Table 4. The recovery rate of this method in actual samples is within the acceptable range of 87.97–97.06%, which proves that this method has high accuracy and sensitivity in real environment [53].

After mixing the two pesticides in accordance with the volume ratio of 1:1, the apple surface was treated with mixed pesticide solution and then SERS detection was conducted. The characteristic peaks of the spectrum obtained by the detection of apple surface samples were consistent with the characteristic peaks of the standard solution. The main characteristic peaks of the two pesticides were reflected in the SERS spectrum of mixed solution. The results were shown in Figure 8. The characteristic peaks of CPF and 2,4-D can still be observed until the concentration was 0.001 mg L^−1^, such as 341, 392, 613, and 675 cm^−1^ Raman characteristic peaks. The result is consistent with the standard solution, indicating that the method is effective and could be used for simultaneous determination of mixed pesticide residues on apple surface.

As the mixed pesticide may not be mixed in a 1:1 ratio in actual situations, we mixed CPF and 2,4-D with different proportions (1:1, 1:2, 1:3, 1:4, 1:5, 1:6, 6:1, 5:1, 4:1, 3:1, 2:1, 1:1 CPF/2,4-D) and then dropped it onto an apple surface for testing. Figure 9 shows the spectra of CPF and 2,4-D with different mixing ratios on apple surface. As labelled, the 341, 621, and 675 cm^−1^ bands of the spectra belong to CPF, and the 392, 852, and 1587 cm^−1^ bands are derived from 2,4-D. Figure 9a shows that when the CPF volume ratio is one, the calculated CPF concentration range is 0.14–0.5 mg L^−1^, and the intensity of the characteristic peak at 675 cm^−1^ diminishes as the CPF concentration decreases. Figure 9b shows that when the volume ratio of 2,4-D is one, the calculated concentration range of 2,4-D is 0.14–0.5 mg L^−1^, and the intensity of the characteristic peak at 392 cm^−1^ decreases as the concentration of 2,4-D decreases. The characteristic peaks of the mixed pesticide spectra are sharp and clear. The above results show that the SERS have the ability to perform the high-sensitivity detection of individual and mixed pesticide residues in samples.

Table 5 showed the linear equations with single variable between Raman characteristic peak intensity and concentration of the mixed pesticides. When the concentration range of CPF in the mixed pesticide was 0.14–0.5 mg L^−1^, the peak intensity of the corresponding Raman characteristic peak also weakened, presenting a certain linear relationship when the 2,4-D ratio is one, respectively. The relationship between the intensity of the characteristic peak and the concentration of the sample was shown in Figure 10. Relationships of intensities of the SERS characteristic peak at (a) 675 cm^−1^, (b) 392 cm^−1^, and concentrations of CPF, 2,4-D were shown in Figure 10. By calculating the concentration of single pesticide in the mixed pesticide, it was found that the higher the concentration of pesticide, the stronger the peak strength of Raman characteristic peak, and there is a certain linear relationship.

The results showed the measured could accurately qualify the residue of mixed pesticide on apple surface. The measured spectrum was consistent with that of the standard solution, and the linear range of the mixed pesticide was 0.001–1000 mg L^−1^. In agricultural production and daily life, this method was suitable for pesticide residues on the surface of complete apples.

The study for individual and mixed pesticides on apple surface showed that CPF and 2,4-D residues on apple surface could be quantified. The limits of detection are 1.28 × 10^−9^ mol L^−1^ for CPF and 2.47 × 10^−10^ mol L^−1^ for 2,4-D. The simultaneous detection of multiple pesticides on apple surface may have potential to be a tool for real-word pesticide testing. The range of CPF and 2,4-D concentration measured in our study was consistent with previous reports regarding the need for CPF and 2,4-D detection. In the determination of CPF in the extract solution from apple with the limits of detection of 10 ng mL^−1^ [50]. In the determination of 2,4-D in spiked tea and milk samples with the limits of detection of 0.11 ng mL^−1^ [54].

### 3.4. Reproducibility of the SERS Measurement

Reproducibility is important for quantitative analysis, especially when SERS was used to detect pesticide residues in agricultural production and daily life. SERS spectra was examined for the mixed solution of 1 mg L^−1^. Mixed pesticide residues were detected on apple surface with Silver colloid prepared by reduction of silver nitrate with hydroxylamine hydrochloride. Figure 11 showed the results of 400 repeated SERS measurements by point-to-point SERS mapping over a 20 μm × 20 μm area on an apple surface sample. The SERS spectra presented satisfying reproducibility, and SERS mapping images showed good uniformity at 675 cm^−1^ and 392 cm^−1^.

## 4. Conclusions

In this study, silver nanoparticles were prepared by reducing silver nitrate with hydroxylamine hydrochloride as SERS substrate, which was then used to predict CPF and 2,4-D residues on apple surface. Simple sample pretreatment method was adopted to conduct SERS detection of the sample, and linear equations were established respectively according to the relationship between SERS characteristic peak and sample concentration. The determination result of the apple sample was consistent with that of the standard solution. The minimum residual concentrations of CPF and 2,4-D on apple surface were 0.001 mg L^−1^ (2.85 × 10^−9^ mol L^−1^) and 0.0001 mg L^−1^ (4.5 × 10^−10^ mol L^−1^), respectively. The measurement results were far below the criteria for CPF and 2,4-D used for China and EU. Mixed CPF and 2,4-D solution with different ratio were also measured with SERS. Characteristic peaks of both drugs were well presented in the SERS spectra, showing that SERS can be used to accurately detect different contents at the same time. This method can be used to quickly and economically detect mixed CPF and 2,4-D residues on apple surface. Moreover, the method may be easily adapted to varieties of agricultural products.

## Figures and Tables

**Figure 1 foods-11-01089-f001:**
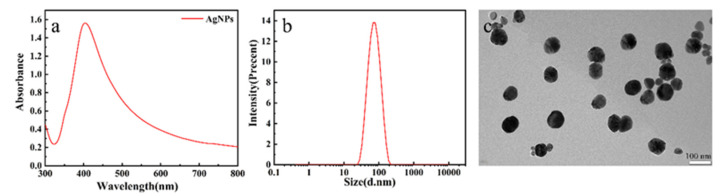
(**a**) UV–vis absorption spectrum of silver nanoparticles. (**b**) The particle size distribution of the silver colloid. (**c**) The TEM images of silver nanoparticles.

**Figure 2 foods-11-01089-f002:**
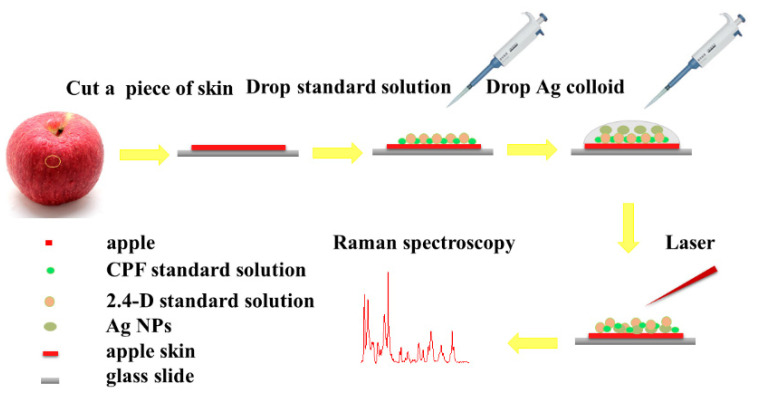
Flowchart of the sample preparation procedure.

**Figure 3 foods-11-01089-f003:**
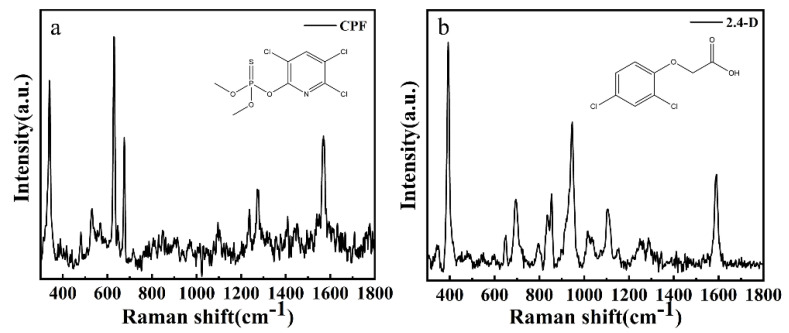
The molecular structure and Raman spectrum of (**a**) CPF and (**b**) 2,4-D.

**Figure 4 foods-11-01089-f004:**
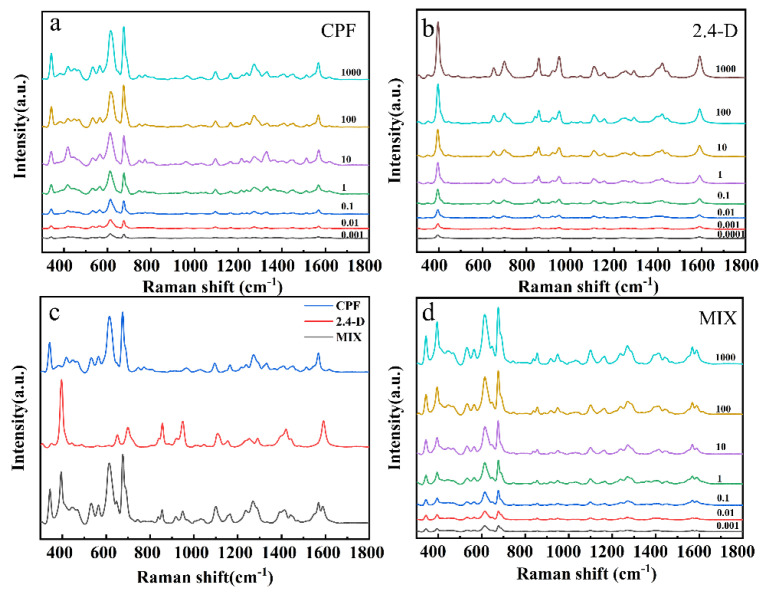
SERS spectra of the (**a**) CPF, (**b**) 2,4-D, (**c**) SERS spectra of the 1 mg L^−1^ CPF, 2,4-D and mixed standard solution, (**d**) mixed standard solution.

**Figure 5 foods-11-01089-f005:**
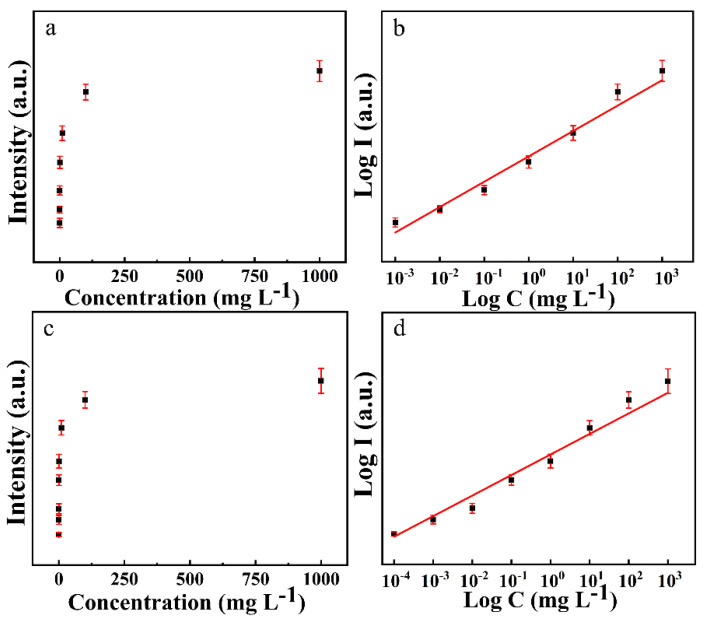
Linear relationship established for CPF and 2,4-D standard solution. Relationships of intensities of the SERS characteristic peak at 675 cm^−1^ and concentrations of CPF shown in linear coordinates (**a**) and log coordinates (**b**). Relationships of intensities of the SERS characteristic peak at 392 cm^−1^ and concentrations of 2,4-D shown in linear coordinates (**c**) and log coordinates (**d**).

**Figure 6 foods-11-01089-f006:**
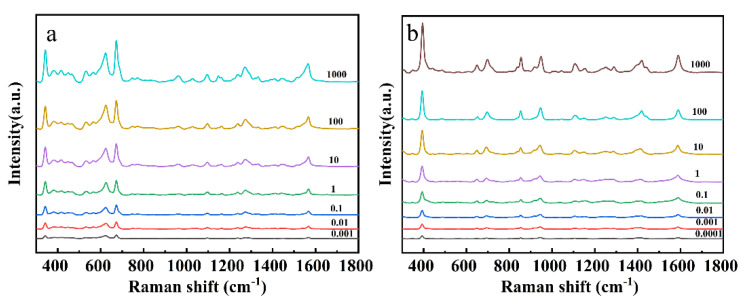
SERS spectra of apple surface samples treated with (**a**) CPF and (**b**) 2,4-D.

**Figure 7 foods-11-01089-f007:**
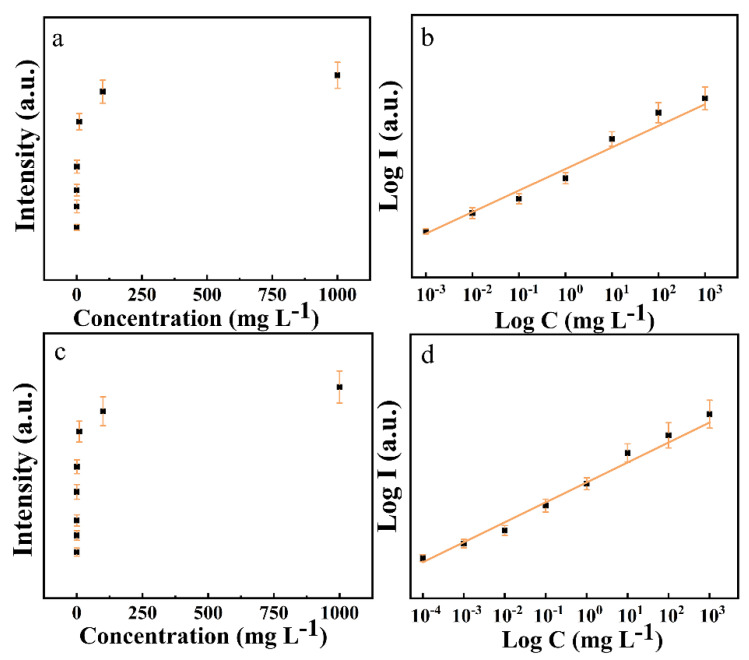
Linear relationship established of apple surface samples treated with (**a**) CPF and (**b**) 2,4-D. Relationships of intensities of the SERS characteristic peak at 675 cm^−1^ and concentrations of apple surface samples treated with CPF shown in linear coordinates (**a**) and log coordinates (**b**). Relationships of intensities of the SERS characteristic peak at 392 cm^−1^ and concentrations of apple surface samples treated with 2,4-D shown in linear coordinates (**c**) and log coordinates (**d**).

**Figure 8 foods-11-01089-f008:**
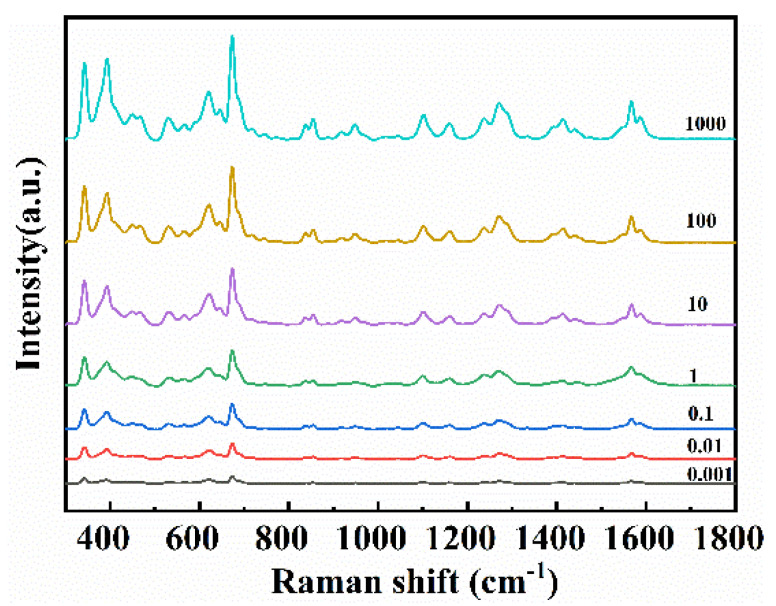
SERS spectra of apple surface samples treated mixed pesticide.

**Figure 9 foods-11-01089-f009:**
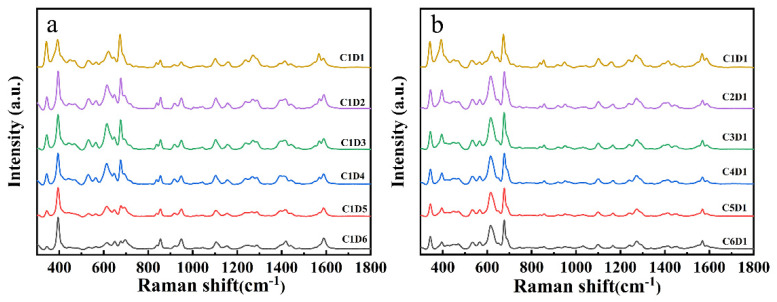
SERS spectra of apple surface samples treated with mixed pesticide of different volume ratio (**a**) CPF volume ratio is one (**b**) 2,4-D volume ratio is one (C1D1 means mixed CPF and 2,4-D at a concentration of 1 mg L^−1^ in 1:1).

**Figure 10 foods-11-01089-f010:**
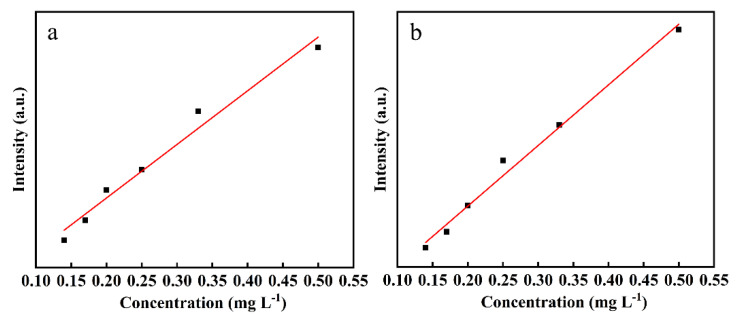
Linear relationship for apple surface samples treated with mixed pesticide. Relationships of intensities of the SERS characteristic peak at (**a**) 675 cm^−1^, (**b**) 392 cm^−1^.

**Figure 11 foods-11-01089-f011:**
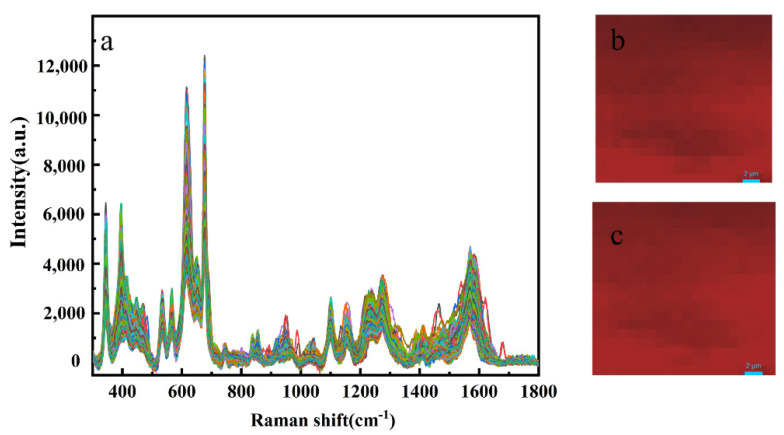
(**a**) SERS spectra of one apple surface sample treated with 1 mg L^−1^ mixed pesticide. (**b**) The SERS mapping image at 675 cm^−1^. (**c**) The SERS mapping image at 392 cm^−1^.

**Table 1 foods-11-01089-t001:** Assignments to CPF and 2,4-D Raman and SERS spectra peaks.

Analyte	Raman/cm^−1^	SERS/cm^−1^	Assignment
CPF	340	341	N-cyclopropyl bending vibration
	411	395	P–O–C stretch
	631	613	P=S
	678	675	P=S
	970	962	Cl-ring wagging
	1103	1094	P–O–C stretch
	1240	1164	Cl-ring, δ(C_H)
	1278	1269	Cl-ring, δ(C_H), νas(C=C)
	1409	1406	Cl-ring, ν(C_N), δ(C_H)
	1455	1448	Cl-ring, ν(C=C)
	1573	1567	Ring stretching
2,4-D	386	392	δ(COC) + δ(CCl)
	662	646	υ(CC)ring + υ(C-Cl)
	709	696	δ(COO−) + υ(CC)ring
	860	855	υ(CC) ring + υ(C-O) + υ(C-Cl)
	956	945	υ(C- COO−)
	1092	1101	υ(CC)ring + υ(CO) + δ(CH)ring
	1428	1415	υs (COO−) + υ(CC) + ω(CH2)
	1590	1590	υas (COO−) + υ(CC)ring

**Table 2 foods-11-01089-t002:** Linear relationship between concentrations of CPF and 2,4-D standard solution and Raman intensities at characteristic peaks.

Analyte	Peaks (cm^−1^)	Linear Equation	R^2^
CPF	342	y = 0.215∗x + 3.70	0.96
	613	y = 0.141∗x + 4.15	0.95
	675	y = 0.184∗x + 4.08	0.97
2,4-D	392	y = 0.185∗x + 4.09	0.98
	852	y = 0.189∗x + 3.76	0.95
	1587	y = 0.167∗x + 3.68	0.96

x = log C; y = log I.

**Table 3 foods-11-01089-t003:** Linear relationship between concentrations of CPF and 2,4-D on apple surfaces and Ra-man intensities.

Analyte	Peaks (cm^−1^)	Linear Equation	R^2^
CPF	341	y = 0.159∗x + 3.91	0.96
	621	y = 0.158∗x + 3.88	0.96
	675	y = 0.155∗x + 3.97	0.98
2,4-D	392	y = 0.175∗x + 4.01	0.98
	852	y = 0.174∗x + 3.57	0.96
	1587	y = 0.165∗x + 3.61	0.97

x = log C; y = log I.

**Table 4 foods-11-01089-t004:** Recovery results of CPF and 2,4-D on apple surface sample.

Analyte	Added Concentration (mg L^−1^)	Detected Concentration (mg L^−1^)	Recovery Rate (%)
CPF	100	96.210	96.21
	1	0.8930	89.30
	0.01	0.0093	93.35
2,4-D	100	97.060	97.06
	1	0.8797	87.97
	0.01	0.0089	89.45

**Table 5 foods-11-01089-t005:** Linear relationship between concentrations of mixed pesticide and Raman intensities on apple surfaces.

Peaks (cm^−1^)	Linear Equation	R^2^
392	y = −373.664∗x + 23792.31	0.98
675	y = −365.346∗x + 33583.38	0.96

x = concentration; y = intensity.

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
