# Peer review of "Rapid Determination of Mixed Pesticide Residues on Apple Surfaces by Surface-Enhanced Raman Spectroscopy"

_foods, 2022, doi:10.3390/foods11081089_

Round 1

Reviewer 1 Report

Dear Authors,

The submitted manuscript (Manuscript Number: foods-1645199) needs extensive editing and rewriting. Discussion is a not well written. There are also too few references to research by other authors. Moreover, there were some errors and writing format problems. 

The purpose of the paper (Manuscript Number: foods-1645199) is rapid determination of CPF and 2.4-D residues on apple surfaces by surface‑enhanced Raman spectroscopy. The subject of the paper itself is maybe inspiring and interesting to readers, however, the work is not well written. The authors do not refer in discussion to studies of the other ones, there is no reference to not a single literature review.   Under this section there are only some test results obtained. 

What more, there are excerpts in material and methods, which should not be included in this section. They do fit much more to the Introduction. This section should not contain sentences such as “Apple is a   widely ………….”, that are supported by literature. 

Furthermore, why authors focused only on a reference between those two compounds?

And how will they demonstrate that the method can be applied for the detection of the residues of other compounds in various agricultural products? 

The paper contains a large number of stylistic and editorial mistakes. 

Reviewer 2 Report

The manuscript foods-1645199 proposes a surface-enhanced Raman scattering (SERS) based method for pesticide detection. A very important thing to consider is that all performance data associated with a quantitative method is based on the calibration procedure: despite the captivating aspect build up by the authors in the first half of the paper, I feel that for this manuscript is hard to reach the level of Foods. The followings are the main comments of this manuscript:

  • Figure 5, 7 and 10. It is not clear if the error bars are confidence interval or a standard deviation. Please clarify it. More importantly, error bars expressed for each calibration level are not worth very much in a calibration plot. A much better (and more effective) visualisation of the calibration behaviour should include confidence and prediction bands for the whole calibration. Please amend the plots and the pertinent caption/part of the text.
  • linear correlation coefficient (R2), is not useful at all to “demonstrate” how good a calibration is. Most importantly, it says nothing about prediction error, the (almost only) one thing the counts for a calibration. Prediction error and bias from validation experiments should be reported instead to declare the accuracy of the quantitative results.
  • Please also provide experimental Limit of quantitation (LOQ). It is a very important Figure of merit in quantitative application since linearity range for the sensor should properly begin at the LOQ level.

Reviewer 3 Report

The authors reported a study for the rapid determination of mixed pesticide residues on apple surfaces by SERS. Even though it is a very important and interesting subject there are some serious issues regarding this manuscript for the authors to address:

  • Line 30. Best-selling is not a good fitting word for the concept, maybe you can use most-selling.
  • Line 106-116. The researchers purchased the apples from the market and washed them with methanol and deionized water. How do they know the methanol and water mixture will remove all the pesticides since they will place a known pesticide concentration on the apple skin later? Why did they not start with an organic apple and still test them with a reference analysis tool for any pesticide/herbicide residue? Since they already finished their analysis, it needs an explanation why they didn’t use an organic apple and can the methanol and water mixture remove all the pesticides. They should add some references to show methanol and water mixture remove all the pesticides.
  • Line 119-134. The spectral collection and data analysis section should be improved. Authors should provide all the specifications for the Raman analysis. The analysis section should be improved; which software was used? They did PLSR or something else? and why did they choose it? etc.
  • Did the authors do any reference analysis to be sure of the chemical concentrations?

Round 2

Reviewer 1 Report

Dear Authors,

I am satisfied with the responses of authors and corrections made in the manuscript (foods-1645199) titled " Rapid determination of mixed pesticide residues on apple surfaces by surface‑enhanced Raman spectroscopy” in light of comments.
I recommend this manuscript for the publication.

Author Response

We greatly appreciate your thoughtful and valuable reviews of our paper entitled “ Rapid determination of mixed pesticide residues on apple surfaces by surface‑enhanced Raman spectroscopy” (Manuscript Number: foods-1645199).

We thank you again for your advice, which is very important. I have found deficiencies in my current work and will make improvements in the future work.

Reviewer 2 Report

I appreciated the efforts of the authors to improve the manuscript.

My recommendation is to accept the revised manuscript.

Author Response

(The authors gave the same response as above.)

Reviewer 3 Report

Authors made the appropriate corrections.

Author Response

We greatly appreciate your thoughtful and valuable reviews of our paper entitled “ Rapid determination of mixed pesticide residues on apple surfaces by surface‑enhanced Raman spectroscopy” (Manuscript Number: foods-1645199).

We thank you again for your advice, which is very important. I have found deficiencies in my current work and will make improvements in the future work. Those comments are all valuable and very helpful for revising and improving our paper, as well as the important guiding significance to our researches. We have studied comments carefully and have made correction which we hope meet with approval.